# Recycling of Vps68 depends on retromer and Mvp1/SNX8

Ralf Kölling*

## ABSTRACT

Here, we analyzed the recycling of the Vps55/Vps68 complex in the yeast endocytic pathway. The two proteins seem to form a functional unit. Single deletions of *VPS55* or *VPS68* affected the turnover of the endocytic cargo protein Ste6 to the same degree. The double deletion had no additive effect on Ste6 turnover. Vps55 and Vps68 were dependent on each other for proper cellular localization. Under normal conditions, the sfGFP-tagged proteins localized to punctate structures; but, when the corresponding partner protein was missing, staining of the vacuolar lumen was observed. This suggests that the orphaned proteins are degraded in the vacuole via the multivesicular bodies (MVB) pathway. A tyrosine-based recycling signal was identified in the cytosolic tail of Vps68. This signal completely restored retromer-dependent recycling and function of a Vps10 variant devoid of its own recycling signals. In line with this finding, Vps68 recycling turned out to be dependent on both retromer and Mvp1/SNX8. This finding was unexpected, since Vps55 recycling seems to depend solely on Mvp1. In co-immunoprecipitation experiments, a weak co-immunoprecipitation signal was detected between Mvp1 and the retromer subunit Vps26, indicating a physical association between Mvp1 and retromer.

KEY WORDS: Recycling, Retromer, Sorting nexin, Scramblase

## INTRODUCTION

Cell surface proteins are internalized by endocytosis and are transported via the endocytic pathway to the lysosomes (or vacuoles in yeast) for degradation. Some of the internalized proteins are spared from degradation by recycling to the Golgi or to the plasma membrane (Cullen and Steinberg, 2018). Recycling also occurs for some proteins that are directly transported to endosomes via the trans-Golgi network (TGN) (Cooper and Stevens, 1996; Ghosh et al., 2003). A central role in recycling is played by an evolutionary conserved protein complex originally identified in yeast called retromer (Seaman et al., 1997, 1998). It consists of two parts, the three proteins Vps26, Vps29 and Vps35 involved in cargo recognition and the sorting nexins Vps5 and Vps17. In metazoans usually only the trimeric VPS26-VPS29-VPS35 complex is referred to as retromer (Arighi et al., 2004; Seaman, 2004). Most of the time, retromer works in combination with sorting nexins (SNX), which are able to recognize or induce membrane curvature and promote membrane tubulation. SNX proteins contain a Phox homology (PX)

Institut für Lebensmittelwissenschaft und Biotechnologie, Fg. Hefegenetik und Gärungstechnologie, Universität Hohenheim, 70599 Stuttgart, Germany.

*Author for correspondence (ralf.koelling@uni-hohenheim.de)

 R.K., 0000-0002-8024-5686

domain, which recognizes phosphoinositides in the target membrane (Cheever et al., 2001; Yu and Lemmon, 2001) and a banana-shaped Bin/Amphiphysin/RVS (BAR) domain involved in sculpting the membrane (van Weering et al., 2010). There are eight SNX proteins in yeast (Shortill et al., 2022). The structure of the retromer complex assembled onto membrane tubules together with the sorting nexin Vps5 was determined by cryo-electron tomography (Kovtun et al., 2018). Retromer forms a dimeric arch-like structure with Vps35 'legs'. Vps26 connects the Vps35 arches to Vps5, which coats the membrane. Vps29 is localized at the tip of the arches.

Here, we investigated the recycling of two yeast proteins, Vps55 and Vps68, involved in the transport of cargo proteins to the vacuole (Belgareh-Touze et al., 2002; Bonangelino et al., 2002). Both proteins are similar in their structure. They are small proteins with four predicted transmembrane domains. Vps55 is the yeast ortholog of human Endospanin-1 (Endo1), a negative regulator of cell surface expression of the leptin receptor (OB-R) (Seron et al., 2011; Vauthier et al., 2017). Vps55 and Vps68 form a complex (Schluter et al., 2008). Evidence has been presented that Vps55 recycling is mediated by the sorting nexin Mvp1/SNX8, independent of retromer (Suzuki et al., 2021). In another study, a certain degree of redundancy between retromer-dependent and Mvp1-dependent recycling of Vps55 was noted (Bean et al., 2017).

Recently, we found that Vps68 physically interacts with the endosomal sorting complex required for transport (ESCRT)-III complex (Alsleben and Kölling, 2022). ESCRT-III is required for the formation of intraluminal vesicles (ILVs) at multivesicular bodies (MVB) (Babst et al., 2002). This tight association suggests that the Vps55/Vps68 complex cooperates with ESCRT-III in ILV formation. The series of events occurring during the process of conversion of early endosomes to MVB and, finally, tethering and fusion with the lysosome/vacuole must be tightly coordinated. For instance, tethering and fusion of the MVB with the vacuole should only occur after ILV formation has been finished. If Vps55/Vps68 is an integral part of the ILV-forming machinery, recycling of the proteins to early endosomes or the Golgi must be prevented until its function is completed. We were thus interested in the mechanisms that control recycling of Vps55/Vps68.

Here, we show that Vps68 contains a tyrosine-based recycling signal in its C-terminal cytosolic tail. This signal completely restored retromer-dependent recycling of the cargo protein Vps10 devoid of its own sorting signals (Suzuki et al., 2019). In contrast to Vps55, the recycling of which appears to rely solely on Mvp1 (Suzuki et al., 2021), Vps68 recycling was dependent on both Mvp1 and retromer. Co-immunoprecipitation (co-IP) experiments suggest that at least a fraction of Mvp1 could associate with retromer.

## RESULTS

### Localization of Vps55 and Vps68

It has been reported that Vps55 and Vps68 form a complex (Schluter et al., 2008). Therefore, we were interested to see whether the deletion of the two encoding genes leads to the same phenotype.

The effect of the deletions on endocytic trafficking was assessed by following the delivery of the short-lived endocytic cargo protein Ste6 to the vacuole for degradation via the MVB pathway. As an indicator of vacuolar delivery, the half-life of Ste6 was determined by a gal-depletion experiment (Fig. 1). In the wild-type strain, Ste6 was turned over with a half-life of 11 min. As reported previously, Ste6 was moderately stabilized in the Δvps68 mutant (Alsleben and Kölling, 2022) (half-life, 16 min). A similar degree of stabilization was observed in Δvps55 and in the Δvps55 Δvps68 double mutant (half-life, 18 min). Thus, in line with the notion that Vps55 and Vps68 form a complex, no additive effect of the individual deletions was observed.

To be able to detect the intracellular localization of Vps55 and Vps68 by fluorescence microscopy, the proteins were tagged with sfGFP at the N- or C-terminus. The functionality of the constructs was again tested by looking at the turnover of Ste6 in the mutant strains (Fig. S1). Ste6 turnover was unaffected by N-terminal tagging of Vps68; thus, sfGFP-Vps68 seems to be functional. In contrast, all the other variants (sfGFP-Vps55, Vps55-sfGFP and

Vps68-sfGFP) showed the same degree of stabilization as the Δvps55 or Δvps68 deletion and are thus non-functional.

The intracellular distribution of the variants was examined by fluorescence microscopy (Fig. 2). Several cytosolic dots and occasional staining of the vacuolar membrane were observed. Staining of the vacuolar membrane was taken as an indication that recycling of the proteins is compromised. The numbers of dots and vacuolar rings were quantified (Fig. S2). With the sfGFP-Vps68 variant, expressed from the VPS68 promoter at its chromosomal locus, a small number of endosomal dots with on average four to five dots per cell was observed. No vacuolar rings were detected. In previous experiments, we expressed sfGFP-Vps68 from the SNF7 promoter, which amounts to a moderate fivefold overexpression of Vps68. In this case, about 30% of the cells displayed vacuolar rings in addition to the cytosolic dots. As noted previously for the endosomal SNARE protein Pep12, overexpression may shift the intracellular distribution of endosomal proteins towards the vacuole (Black and Pelham, 2000). When Vps68 was tagged at its C-terminus, the distribution of endosomal dots was shifted to smaller numbers with a peak at one to two dots per cell. In addition, about 50% of the cells displayed vacuolar rings. This suggests that recycling of Vps68 is compromised by C-terminal tagging. For sfGFP-Vps55, the distribution of dots was similar to the distribution of Vps68-sfGFP. An interesting phenotype was observed with C-terminally tagged Vps55. With this variant, eight or more dots per cell, with a maximum at seven dots and virtually no vacuolar rings, were observed. This could be a sign of enhanced recycling of the protein.

### Vps55 and Vps68 depend on each other for proper localization

Recycling of Vps55 has been studied in detail (Suzuki et al., 2021). Since Vps55 and Vps68 form a complex, we wanted to know which of the two components is responsible for the proper localization of the complex. To test this, the localization of the proteins was examined in mutants, where the corresponding complex partner was deleted (Fig. 2). Under these conditions, both Vps55-sfGFP and sfGFP-Vps68 no longer formed endosomal dots but instead accumulated in the lumen of the vacuole as marked with the dye FM4-64. This suggests that the orphaned subunits of the complex become a substrate for the MVB pathway and are finally degraded in the vacuole. In line with this finding, it has been reported previously that Vps55 is ubiquitinated and degraded in the vacuole upon overexpression (Suzuki et al., 2021). These results clearly show that the two proteins are dependent on each other for proper localization.

### The C-terminus of Vps68 contains a recycling signal

Next, we were interested in defining the sorting signals in Vps55/Vps68 that are required for retrieval of the complex. For this, we made use of the well-studied retromer cargo protein Vps10, a sorting receptor for vacuolar hydrolases, which cycles between endosomes and the Golgi (Cooper and Stevens, 1996; Seaman et al., 1997, 1998). Vps10 tagged with sfGFP at its C-terminus was localized to a handful of intracellular dots (Fig. 3). Two sorting signals had been identified within Vps10 (Suzuki et al., 2019). Deletion of these two signals (Vps10Δ) shifted the localization completely to the vacuolar membrane, due to defective retrieval to early endosomes or the Golgi. The C-terminal sequences downstream from the identified sorting signals [amino acids (aa) 1497-1579] are highly conserved among Vps10 homologs (Vth1, Vth2, Ysn1). They constitute an intrinsically disordered region (IDR). Deletion of this region has only a slight impact on CPY sorting, but seems to affect the

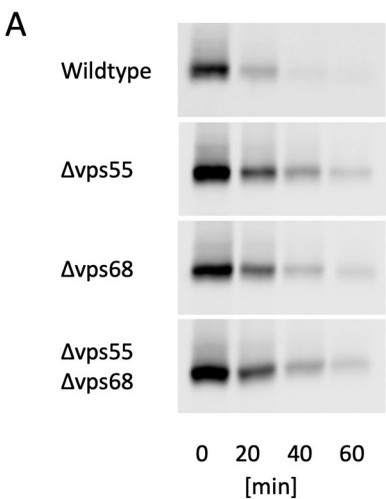

**A**

Wildtype

Δvps55

Δvps68

Δvps55
Δvps68

0   20   40   60
[min]

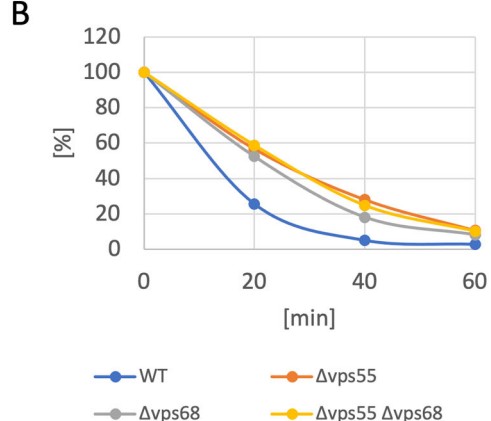

**B**

**Fig. 1. Ste6 turnover in VPS55 and VPS68 mutants.** Ste6 turnover was examined by a gal-depletion experiment. Cells were first grown in YP-Gal medium and were then shifted to YPD. Cell aliquots were taken at 20 min intervals after a 20-min preincubation in glucose medium and analyzed for Ste6 by western blotting. (A) From top to bottom: RKY3319 [wild type (WT)], RKY3379 (Δvps55), RKY3320 (Δvps68), RKY3435 (Δvps55 Δvps68). (B) Quantification of the western blot signals by ImageJ.

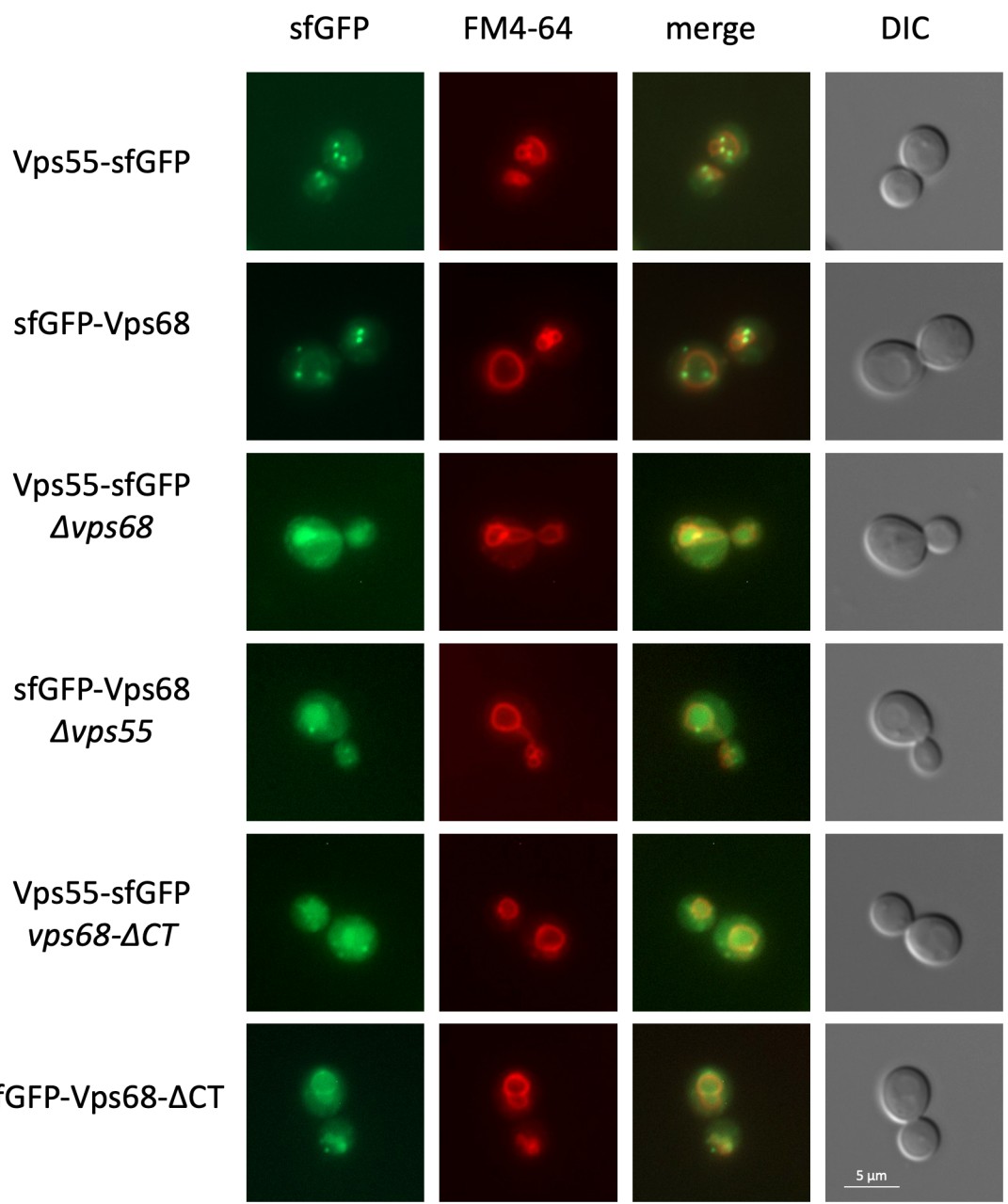

**Fig. 2. Vps55 and Vps68 depend on each other for correct cellular localization.** The localization of Vps55-sfGFP and sfGFP-Vps68 was examined by fluorescence microscopy in wild-type and mutant backgrounds. Cells were additionally labeled with the dye FM4-64 to mark the position of the vacuoles. Strains from top to bottom: RKY3895 (Vps55-sfGFP, WT), RKY3824 (sfGFP-Vps68, WT), RKY3856 (Vps55-sfGFP, Δvps68), RKY3871 (sfGFP-Vps68, Δvps55), RKY3854 (Vps55-sfGFP, vps68-ΔCT), RKY3877 (sfGFP-Vps68-ΔCT). Panels from left to right: sfGFP fluorescence, FM4-64 staining, merged images, differential interference contrast (DIC) images.

stability of the protein (Cereghino et al., 1995). This region is retained in the Vps10Δ variant. We now used Vps10Δ as a reporter for the detection of Vps55/Vps68 sorting signals. For both Vps55 and Vps68 four transmembrane spanning domains (TMDs) are predicted from the sequence by DeepTMHMM (https://dtu.biolib. com/DeepTMHMM). For Vps68, we presented data pointing to an unusual membrane topology with only two TMDs (Alsleben and Kölling, 2022). However, we now found that the hydrophobic 3HA-tag used in the reporter construct interfered with the outcome of the experiment. We thus believe now that Vps55 and Vps68 most likely have a conventional membrane topology, as suggested by others (Seron et al., 2011; Suzuki et al., 2021). Based on this membrane

topology, only three segments of Vps55 and Vps68, the N- and C-termini and the loop between TMD 2 and 3, are exposed to the cytosol. These sequences were inserted into our Vps10Δ reporter construct (Fig. 4). Five of the six segments had no effect on the localization of the reporter protein, which was still exclusively found at the vacuolar membrane. Only the C-terminal tail of Vps68 was able to restore recycling of Vps10Δ. The Vps68 C-terminus consists of the sequence QNVEDE**YS**YS**YS**LTL containing two tyrosine residues (highlighted by bold print). Tyrosine-based sorting signals have been implicated in the sorting of membrane proteins to various intracellular compartments (Bonifacino and Dell'Angelica, 1999). To see if the two tyrosine residues in the

A

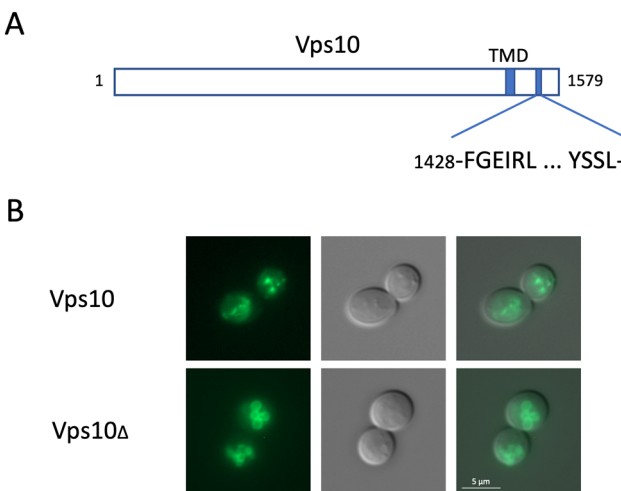

Vps10

1 ▯ TMD ▯ 1579

1428-FGEIRL … YSSL-1495

B

Vps10

Vps10Δ

5 μm

**Fig. 3. Vps10 reporter construct for the detection of recycling signals.** (A) Schematic view of Vps10. Vps10 consists of a large N-terminal luminal domain, a single transmembrane domain (TMD) and a cytosolic tail containing two sorting signals (aa 1428-1495), which are deleted in Vps10Δ. (B) Fluorescence images of sfGFP-tagged Vps10 variants, expressed from single-copy plasmids transformed into JD52. Top, wild-type Vps10 (pRK2067); bottom, Vps10Δ with sorting signals deleted (pRK2070). From left to right: sfGFP staining, DIC picture, merged picture.

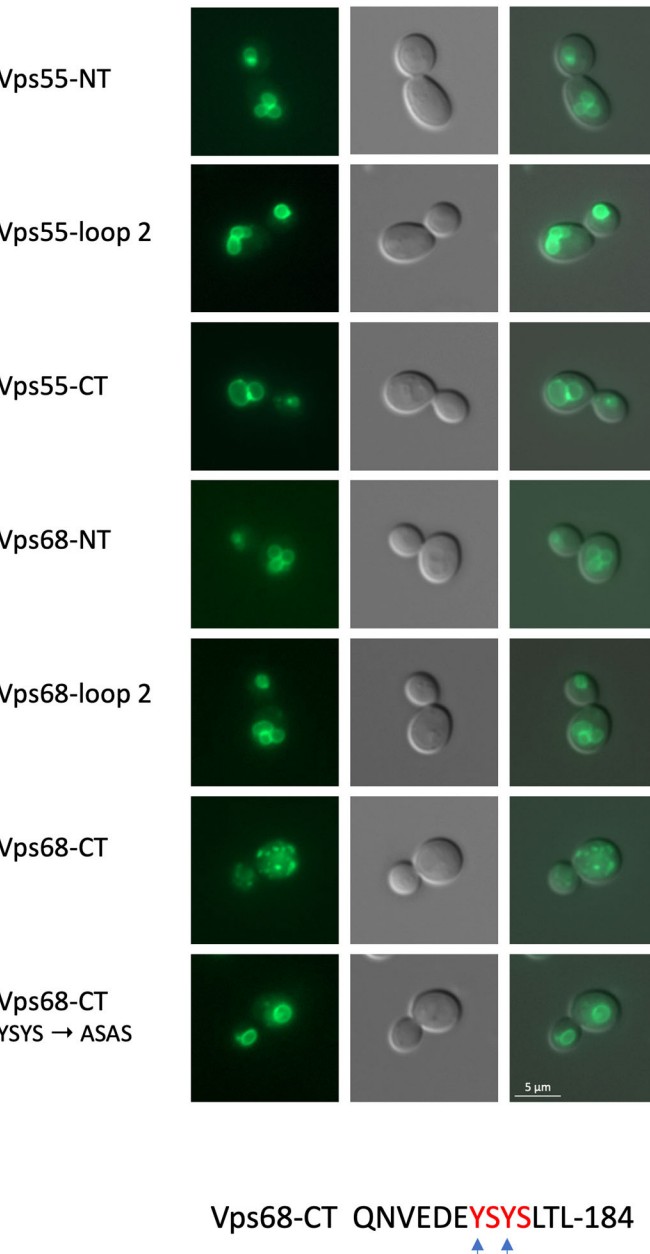

Vps55-NT

Vps55-loop 2

Vps55-CT

Vps68-NT

Vps68-loop 2

Vps68-CT

Vps68-CT
YSYS → ASAS

5 μm

Vps68-CT QNVEDEYSYSLTL-184
↑ ↑
A A

**Fig. 4. The C-terminus of Vps68 contains a tyrosine-based recycling signal.** Vps55 and Vps68 fragments were inserted into the Vps10Δ reporter, and the localization of the sfGFP-tagged variants, expressed from single-copy plasmids in strain JD52, was examined by fluorescence microscopy. Fragments inserted into Vps10Δ, from top to bottom: pRK2076 (Vps55 N-terminus), pRK2083 (Vps55 loop 2, between TM2 and TM3), pRK2077 (Vps55 C-terminus), pRK2074 (Vps68 N-terminus), pRK2084 (Vps68 loop 2, between TM2 and TM3), pRK2075 (Vps68 C-terminus), pRK2085 (Vps68 C-terminus with tyrosine residues mutated to alanine). From left to right: sfGFP staining, DIC picture, merged picture. At the bottom, the Vps68 C-terminal sequence with the alanine ('A') mutations is depicted.

Vps68 C-tail are crucial for recycling, they were mutated to alanine. Indeed, the YSYS → ASAS mutation abrogated recycling of Vps10.

To see if the replacement of the Vps10 sorting signals by the Vps68 YSYS signal gives rise to a functional protein, the sorting of an CPY-invertase chimera to the vacuole was examined. Lack of Vps10 sorting receptor activity leads to missorting of CPY-invertase to the culture medium (Bankaitis et al., 1986; Rothman and Stevens, 1986). The extracellular invertase activity can be detected directly on agar plates by an overlay assay, which leads to the development of a green-blue color (Darsow et al., 2000). A Δvps10 strain was transformed with the single-copy CPY-invertase reporter plasmid and with a single-copy plasmid expressing different Vps10-sfGFP variants or with an empty vector (Fig. 5). Transformation with the empty vector gave rise to a strong staining, due to missorting of CPY-invertase to the culture medium. The plasmid expressing wild-type Vps10-sfGFP in contrast reduced the external invertase activity to background levels (similar to the staining observed with the wild-type strain transformed with the reporter plasmid). The Vps10Δ variant proved to be non-functional. The cells expressing this variant looked like the vector control. The cells expressing the Vps10Δ variant with the Vps68 C-terminus, however, were indistinguishable from the cell expressing wild-type Vps10. This shows that the Vps68 C-terminus can functionally replace the native Vps10 sorting signals.

Next, the impact of the Vps68 recycling signal on the localization of Vps55 and Vps68 was tested in its native context (Fig. 2). The deletion of the C-terminal signal of Vps68 led to a mislocalization of Vps55-sfGFP similar to the complete deletion of *VPS68*, likewise sfGFP-Vps68 with the C-terminal deletion was mislocalized to the vacuolar lumen. We have thus presented evidence that the C-terminus of Vps68 contains a tyrosine-based sorting signal affecting the localization of the Vps55-Vps68 complex.

### The Vps68 recycling signal mediates retromer-dependent recycling of Vps10

Recycling of Vps10 is retromer dependent (Suzuki et al., 2019), while recycling of Vps55 was shown to depend exclusively on the

sorting nexin Mvp1 (Suzuki et al., 2021). To see how the YSYS signal of Vps68 behaves in the context of Vps10, the corresponding reporter construct was expressed in the sorting nexin mutants Δsnx3, Δsnx4, Δmvp1 and in the retromer mutant Δvps35 (Fig. 6). Surprisingly, we found that the construct with the Vps68 signal behaved like wild-type Vps10. Recycling was blocked in Δsnx3

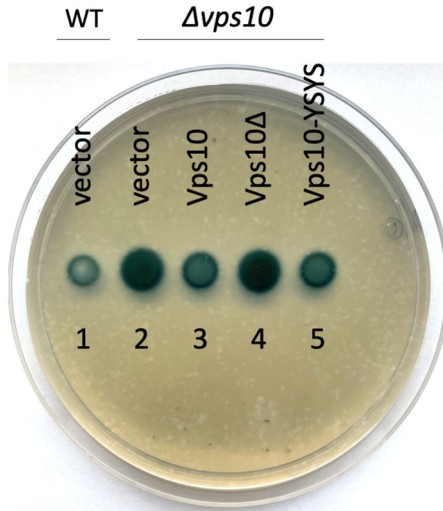

**Fig. 5. The C-terminal signal of Vps68 restores Vps10 function.**
Overnight cultures were spotted onto SD/CAS plates containing 2%
galactose. The plates were incubated for 1 day at 30°C, and then external
invertase activity was measured by an invertase overlay assay. External
invertase activity gives rise to a green-blue color. (1) RKY3977 (*VPS10
Δsuc2*) transformed with the reporter plasmid pRK2196 (*PRC1-SUC2*) and
the empty vector YCplac22. (2-5) RKY3978 (*Δvps10 Δsuc2*)/pRK2196
transformed with (2) YCplac22, (3) pRK2067 (*VPS10-sfGFP*), (4) pRK2070
(*VPS10Δ-sfGFP*), (5) pRK2075 (*VPS10Δ-YSYS-sfGFP*). (2,4) These spots
appear larger due to the diffusion of the colored reaction product into the
surrounding medium. All spots were initially of the same size. Two
independent transformants each were tested.

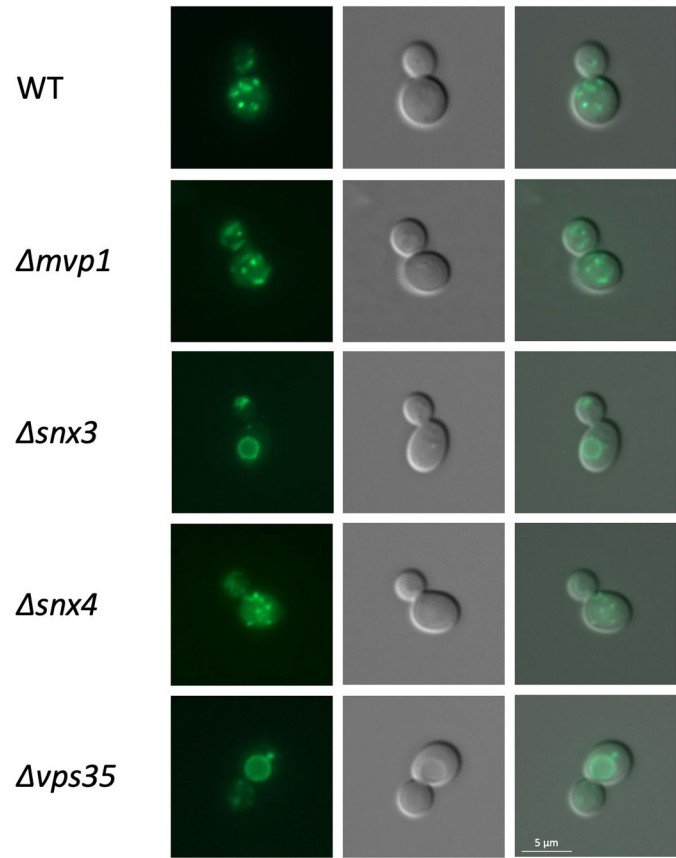

**Fig. 6. Intracellular distribution of the Vps10Δ/Vps68 CT chimera in
different sorting nexin and retromer mutants.** Plasmid pRK2075,
expressing the Vps10Δ reporter with the Vps68 C-terminus, was transformed
into different strains. From top to bottom: BY4741 (WT), Y01654 (*Δsnx3*),
Y01386 (*Δsnx4*), RKY2074 (*Δvps35*), Y06533 (*Δmvp1*). From left to right:
sfGFP staining, DIC picture, merged picture.

and in *Δvps35*, but was unaffected by *Δsnx4* and *Δmvp1*. This
indicates that the Vps68 recycling signal is a bona fide retromer
signal. This finding was unexpected, since Vps55 recycling is solely
dependent on Mvp1, and since Vps55 and Vps68 appear to form a
complex, one would expect that they share the same sorting nexin
requirements. To test this prediction, the sorting nexin requirement
of Vps55 and Vps68 was examined directly.

### Recycling of Vps68 depends on both Mvp1 and Vps35

The intracellular distribution of Vps55-sfGFP and sfGFP-Vps68
was examined in the wild-type and in *Δmvp1* and *Δvps35* mutants
(Fig. 7). With respect to Vps55-sfGFP, we could confirm the
previously reported results (Suzuki et al., 2021). In the wild type, we
observed a handful of endosomal dots and no vacuolar rings. In the
*Δmvp1* mutant, all cells contained vacuolar rings with one or
two dots at the vacuolar membrane. The staining pattern in the
*Δvps35* mutant was indistinguishable from wild type. The results
with sfGFP-Vps68, however, were different. While there were
only endosomal dots and no vacuolar rings visible in the wild-type
strain, deletion of *MVP1* and *VPS35* led to the same phenotype,
accumulation of sfGFP-Vps68 in vacuolar rings with one or
two associated dots. To exclude that the recycling defect observed
in the *Δvps35* mutant was due to secondary effects in the strain
background, a complementation experiment was performed. The
strain was transformed with a single-copy plasmid expressing
Vps35 or with an empty vector. As can be seen in Fig. 7, the *VPS35*
plasmid completely complemented the recycling defect and restored
a wild-type staining pattern. From this, we conclude that Vps68
recycling depends on both Mvp1 and retromer.

Our results are most easily explained by the assumption that
Mvp1 and retromer form a complex. Retromer in yeast is usually

associated with the sorting nexins Vps5 and Vps17. A retromer
complex containing Mvp1 has not been demonstrated so far.
To explore this possibility, a co-IP experiment was performed.
Mvp1 was tagged with sfGFP at its C-terminus, and the retromer
subunit Vps26 was tagged with a 13myc tag also at its C-terminus.
C-terminally tagged Vps26 was functional, while N-terminally
tagged Vps26 or Vps35 turned out to be non-functional. We were
not able to obtain a C-terminally tagged Vps35 variant. Vps26-
13myc was immunoprecipitated from cell extracts with anti-myc
antibodies and the precipitates were examined for co-IP of Mvp1-
sfGFP. As can be seen in Fig. 8A, a weak co-IP signal was
observed (0.1% of total protein co-precipitated). This signal
was specific, because it was only obtained in the presence of
antibodies and in a strain carrying tagged Vps26. As a control, an
immunoprecipitation with the canonical retromer component Vps5-
sfGFP was performed (Fig. 8B). As reported previously (Seaman
et al., 1998; Suzuki et al., 2019), a robust co-IP signal between
Vps26 and Vps5 was obtained (4.5% of total protein co-
precipitated), which was about 50 times stronger than the signal
between Vps26 and Mvp1.

### DISCUSSION

Here, we investigated the recycling of Vps55 and Vps68 from
endosomes. In a previous study, the recycling of Vps55 had been
examined in detail (Suzuki et al., 2021). The conclusion from this

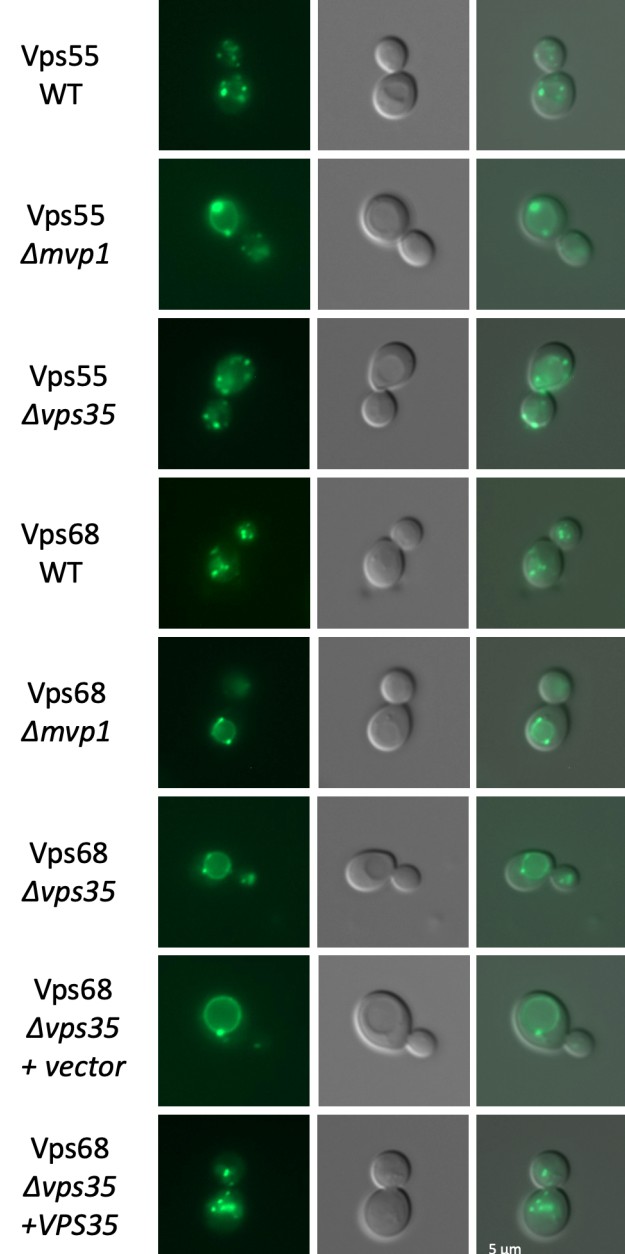

**Fig. 7. Effect of *MVP1* and *VPS35* deletion on Vps55 and Vps68 localization.** The localization of Vps55-sfGFP and sfGFP-Vps68 was examined by fluorescence microscopy in wild-type and mutant backgrounds. Strains from top to bottom: RKY3577 (Vps55-sfGFP, WT), RKY3855 (Vps55-sfGFP, Δ*mvp1*), RKY3597 (Vps55-sfGFP, Δ*vps35*), RKY3824 (sfGFP-Vps68, WT), RKY3853 (sfGFP-Vps68, Δ*mvp1*), RKY3852 (sfGFP-Vps68, Δ*vps35*), RKY3852 (sfGFP-Vps68, Δ*vps35*) transformed with the vector YCplac33, RKY3852 (sfGFP-Vps68, Δ*vps35*) transformed with the *VPS35* plasmid pRK2176. Panels from left to right: sfGFP fluorescence, DIC image, merged image.

study was that recycling of Vps55 relies solely on Mvp1/SNX8 and occurs independently of retromer. Our data confirm this conclusion. Since Vps55 and Vps68 form a complex, one would expect that both proteins share the same sorting nexin requirement. But here, we found that Vps68 recycling was dependent on both Mvp1 and retromer.

What could be the reason for the difference between Vps55 and Vps68? One possibility is that Vps68, in contrast to Vps55, is

recycled by two independent pathways. Indeed, a high-throughput imaging screen suggested a substantial degree of redundancy with respect to the sorting nexin requirements for a number of yeast cargo proteins (Bean et al., 2017). But we consider this possibility unlikely, since both the *MVP1* and the *VPS35* deletion displayed a strong recycling defect and there was no indication of an additive effect of the two mutations. In case of redundant, parallel pathways a partial recycling effect for the single mutations would have been expected. Our data thus suggest that Mvp1 and retromer cooperate in the recycling of Vps68. A similar conclusion for Vps10 sorting was reached in a previous study, although a direct physical interaction between Mvp1 and retromer could not be demonstrated (Chi et al., 2014). In the study by Suzuki et al. (2021), attempts to prove an interaction between Mvp1 and retromer by co-IP were also unsuccessful.

Here, for the first time, we were able to detect a co-IP signal between Mvp1 and a retromer subunit. However, since this signal is much weaker than the signal obtained with the canonical sorting nexin Vps5, it remains an open question whether the observed interaction between retromer and Mvp1 is direct or indirect.

In this context, it is important to define what is meant by 'retromer'. In metazoans, retromer refers to the trimer consisting of VPS26, VPS29 and VPS35 (often called the 'cargo recognition complex' or CSC), while in yeast the two sorting nexins Vps5 and Vps17 are considered to be an integral part of retromer. In this study, we are using the term 'retromer' in the metazoan sense. There is indeed evidence that yeast retromer forms complexes without Vps5 and Vps17. Curiously, deletions of the genes coding for the retromer trimer subunits have different phenotypes than the deletions of the genes coding for the sorting nexins Vps5 and Vps17. Loss of Vps5 or Vps17 leads to a fragmented vacuole phenotype, whereas loss of the retromer trimer subunits gives rise to a normal vacuolar morphology (Raymond et al., 1992). This discrepancy could be explained by the finding that the retromer trimer is able to form a complex with the PROPPINs Atg18 and Atg21 (Courtellemont et al., 2022; Marquardt et al., 2023). Lack of competition with the sorting nexins for retromer binding leads to overactivity of the retromer-PROPPIN complex, which mediates vacuole fission, resulting in hyper-fragmentation of the vacuole.

But why then does Vps68 require both Mvp1 and retromer, while Vps55 is only dependent on Mvp1? The simplest interpretation is that both Vps55 and Vps68 associate with recycling tubules generated by Mvp1. In this scenario, Vps55 would directly bind to Mvp1, as suggested by Suzuki et al. (2021), whereas Vps68 would additionally require the assistance of retromer for binding. This would also mean that both proteins are recycled individually and not as a complex.

Binding of Vps68 to retromer could be mediated by the tyrosine-based sorting signal ('YSYS') that we identified in the Vps68 C-terminus. Vps10, a member of the sortilin receptor family and a well-studied retromer substrate, contains a similar recycling signal 'YSSL' (Cooper and Stevens, 1996). However, it has been reported that this motif alone is not sufficient for efficient retrieval of Vps10 and that an additional motif is necessary for maximum recycling (Suzuki et al., 2019). But here we showed that the YSYS sequence of Vps68 alone is able to fully restore retromer-dependent recycling to a Vps10 variant, in which both ('bipartite') recycling motifs were deleted. This suggests that the YSYS sequence promotes stronger binding to retromer than the YSSL sequence, since it does not require assistance by the second motif.

Tyrosine-based sorting signals with the consensus sequence YXXØ ('Ø'=amino acid with a bulky hydrophobic side chain) have

Biology Open

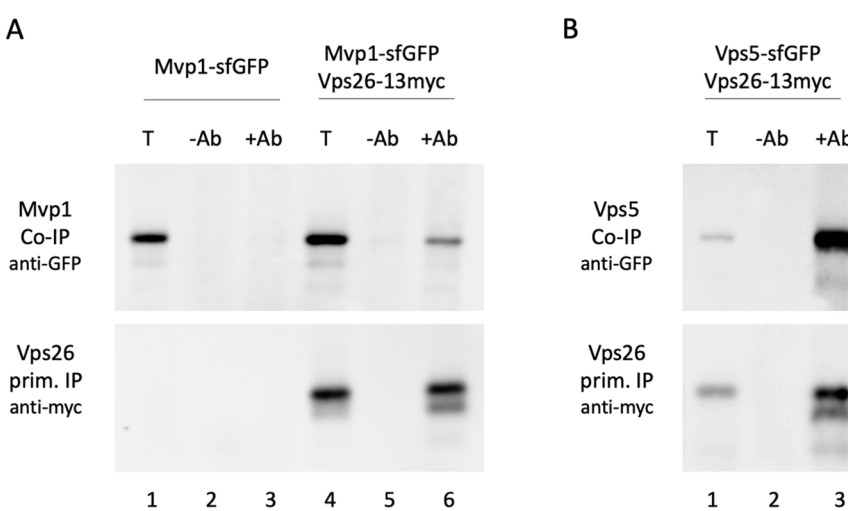

**Fig. 8. Co-immunoprecipitation of Mvp1 or Vps5 by Vps26.** (A) Vps26-13myc was immunoprecipitated from cell extracts of RKY3941 (Mvp1-sfGFP, Vps26-13myc) with anti-myc antibodies. As a negative control, RKY3795 (Mvp1-sfGFP) was used, which does not contain tagged Vps26. Lanes 1-3, RKY3795; lanes 4-6, RKY3941; lanes 1, 4, input; lanes 2, 5, mock IP without antibody; lanes 3, 6, IP with anti-myc antibody. (B) Vps26-13myc was immunoprecipitated from cell extracts of RKY3966 (Vps5-sfGFP, Vps26-13myc) with anti-myc antibodies. Lane 1, input; lane 2, mock IP without antibody; lanes 3, IP with anti-myc antibody. Upper panels, co-immunoprecipitation of Vps5-sfGFP, developed with anti-GFP antibodies; lower panels, primary IP of Vps26-13myc, developed with anti-myc antibodies. Input for primary IP=5% of total, input for co-IP=1% of total. The experiment was performed two times.

been widely implicated in the sorting of transmembrane proteins in the endocytic and post-Golgi sorting pathways (Bonifacino and Dell'Angelica, 1999). Ample structural information is available showing that these signals bind to the medium subunits of clathrin adapter complexes. Several yeast proteins with tyrosine-based signals have been implicated in clathrin adapter complex-dependent transport steps. For instance, the two sorting motifs in Sna2 (YSHL and YGSL) are thought to bind to AP-1 and AP-3 complexes (Renard et al., 2010). Also, the YSAV signal in Atg27 has been linked to its AP-3-dependent transport to the yeast vacuole (Segarra et al., 2015). However, the transport itinerary of Atg27 appears to be more complex. It has been shown that, after its AP-3 dependent transport to the vacuole, Atg27 is recycled to the Golgi in a two-step process involving a Snx4-dependent and a retromer-dependent step (Suzuki and Emr, 2018). These data are compatible with the view that the YSAV signal is indeed a retromer signal. Whether clathrin plays a role in retromer-dependent sorting is a matter of debate. It has been proposed that clathrin and retromer act sequentially in the same recycling pathway (Popoff et al., 2007). This notion, however, has been disputed (McGough and Cullen, 2013). In the same vein, the deletion of the gene for the AP-1 subunit Apl2 had no obvious effect on the intracellular localization of sfGFP-tagged Vps10 or Vps68. But still, a contribution of clathrin adapter complexes to retromer-dependent sorting cannot be completely ruled out at the moment.

## MATERIALS AND METHODS
### Yeast strains and media
For Gal depletion experiments, yeast cells were grown overnight to exponential phase in YP-Gal medium (1% yeast extract, 2% peptone, 2% galactose, 0.2% glucose) and were then shifted to YPD medium (1% yeast extract, 2% peptone, 2% glucose). For fluorescence microscopy, cells were grown in synthetic defined/casamino acids (SD/CAS) medium (0.67% yeast nitrogen base, 1% casamino acids, 2% glucose, 50 mg/l uracil and tryptophan) to avoid autofluorescence from the YPD medium. For plasmid selection, tryptophan or uracil were omitted from the medium. The yeast strains used are listed in Table S1. Most of the yeast strains are derived from JD52 (J. Dohmen, Cologne, Germany) by the integration of PCR cassettes into the yeast genome (Longtine et al., 1998). Plasmids used are listed in Table S2.

### Fluorescence microscopy
Yeast cells were grown overnight to exponential phase in SD/CAS medium at 30°C. The yeast cell suspension was applied to concanavalin A-coated slides and imaged with a Zeiss Axio-Imager M1 fluorescence microscope

equipped with an AxioCam MRm camera (Zeiss, Göttingen, Germany). Images were acquired with the Axiovision Software and processed with Photoshop Elements.

### Co-IP
Yeast cells were grown overnight to exponential phase in YPD medium at 30°C. 20 ml cells with an optical density at 600 nm of 1 were harvested and washed in cold phosphate-buffered saline (PBS; 10 mM $NaH_2PO_4$, 150 mM NaCl, pH 7.2). The cells were then resuspended in 200 µl PBS+1 mM phenylmethylsulfonylfluoride (PMSF; Carl Roth, Karlsruhe, Germany, #6367.1) and agitated with glass beads for 5 min at 4°C on a vortex shaker. After addition of 600 µl PBS/PMSF, the mixture was transferred to a new tube and spun at 500 $g$ for 5 min. The supernatant was mixed with dithiobis-succinimidylpropionate (DSP) (Thermo Fisher Scientific, Rockford, IL, USA, #22585) dissolved in dimethylsulfoxide (DMSO) to a final concentration of 3 mM and incubated for 30 min at room temperature (RT) on a rocker. Then 50 mM of Tris-HCl pH 8.0 and 1% TX-100 were added for quenching of the crosslinking reaction and for solubilization of the membranes and incubated for 15 min at RT on a rocker. Next, the mixture was spun at 500 $g$ for 5 min. An aliquot of the supernatant was taken as input control. The rest of the supernatant was split into two parts. To one part 3 µl of antibody (anti-myc 9E10, BioLegend, London, UK, #904401; anti-GFP, Nordic MUbio, Susteren, The Netherlands, #Bii-nGFPab3-100) were added, while the second part acted as the no-antibody control. Both parts were incubated for 1 h at 4°C on a rocker. Then 100 µl of a 20% slurry of protein A sepharose beads CL-4B (Cytiva, Uppsala, Sweden, #17078001) in PBS buffer were added, and incubation was continued for 1 h at 4°C. The protein A beads were washed three-times with 1 ml PBS buffer by spinning them for 1 min at 500 $g$. The beads were finally resuspended in 100 µl sample buffer/PBS and heated to 95°C for 5 min.

### Invertase overlay-assay
The invertase overlay-assay was performed according to Darsow et al. (2000). Overnight cultures grown in SD/CAS with 2% galactose and 0.2% glucose were spotted onto SD/CAS agar plates containing 2% galactose. The cells on the agar plates were overlayed with a mixture of 2 g sucrose, 5 ml 0.1 M sodium acetate pH 5.5, 0.5 ml peroxidase (1 mg/ml, Sigma-Aldrich, #77332), 0.4 ml glucose oxidase (5 mg/ml, Sigma-Aldrich, #G7141), 3 ml o-dianisidine (10 mg/ml, Sigma-Aldrich, #D3252) and 20 ml deionized water and 20 ml 3% heated agar. The plates were developed for about 10-30 min and then photographed.

### Acknowledgements
I thank Thomas Brune, lab technician responsible for lab organization, for his assistance.

### Competing interests
The author declares no competing or financial interests.

Biology Open

**Funding**
 Deposited in PMC for immediate release.

**Data and resource availability**
All relevant data and details of resources can be found within the article and its supplementary information.

**Peer review history**
The peer review history is available online at https://journals.biologists.com/bio/lookup/doi/10.1242/bio.062518.reviewer-comments.pdf

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
