## [Peer Review File · Biology Open]

Recycling of Vps68 depends on retromer and Mvp1/SNX8

Ralf Kölling

DOI: 10.1242/bio.062518

Editor: Catherine L. Jackson

Review timeline

Submission to sister journal:	13 October 2026
Editorial decision at sister journal:	22 October 2025
First revision received at sister journal:	3 February 2026
Editorial decision at sister journal:	7 February 2026
Transfer to Biology Open:	9 February 2026
Editorial decision:	3 March 2026
First revision received:	10 March 2026
Accepted:	11 March 2026

Original submission to sister journal

First decision letter

MS Title: Recycling of Vps68 depends on retromer and Mvp1/SNX8

Author: Ralf Kölling

We have now reached a decision on the above manuscript.

To see the reviewers' reports and a copy of this decision letter, please go to:

As you will see, the reviewers raise a number of substantial criticisms that prevent me from accepting the paper. One of the reviewers was of the opinion that there was not sufficient advance and the manuscript was not suitable for publication in our journal. I believe it will be difficult to address these concerns in a reasonable timeframe and in view of the comments you might prefer to submit elsewhere. However, I think the fairest course of action is to leave open the possibility of addressing all the comments of the reviewers in a completely revised manuscript. The revised manuscript would need to address both the technical concerns (especially questions of quantitation and rigor) but also the question of novelty. It would then be sent to the original reviewers.

If you do choose to submit a revised manuscript in due course, please upload both a 'clean' version of your Word file, along with a highlighted version clearly showing where you have made changes made in the revised manuscript. Please avoid using 'Tracked changes' in Word files as these are lost in PDF conversion.

I should be grateful if you would also provide a point-by-point response detailing how you have dealt with the points raised by the reviewers in the 'Response to Reviewers' box. Please attend to all of the reviewers' comments. If you do not agree with any of their criticisms or suggestions please explain clearly why this is so.

We are committed to maintaining the integrity of the scientific record. Please read our requirements for preparing your figures to avoid a potential delay in the publication process or

rejection on the basis of non-compliance with these guidelines. This guide also includes recommendations on improving figure layout to help reviewers and readers appreciate your data.

For best practice and transparency, and to allow better assessment of the quality of the data and whether the data support the conclusions, we strongly recommend that you:

1. Use graphs that allow the reader to see the true data spread (i.e. box and whisker plots, super plots, etc. See PMID: 32346721 for more information.). If using box-and-whisker plots, please also state what the whiskers represent. See here for helpful information.
2. Use appropriate statistics with the sample size representing biological replicates rather than technical replicates
3. Provide, if using western blots, one additional supplementary figure (as a single PDF or TIFF file). It will not contribute towards your supplementary figure limit. If you provided this at submission and named it 'Blot transparency', please now number this supplementary figure and combine it with the rest of the supplementary material pdf. Please ensure that you cite this figure in your Materials & Methods section, ideally in the western blots paragraph.

Reviewer 1

Comments for the author

This manuscript investigates the localization of the yeast protein Vps68. The author identified a tyrosine-based motif in Vps68 and reported that its localization depends on both Mvp1 and the retromer complex. They further proposed that Mvp1 associates with retromer, presenting this as the first biochemical evidence for a joint function of Mvp1/SNX8 and retromer. Additionally, the study examined the phosphorylation of Vps55 and Vps68. However, I could not identify any major or novel conclusion arising from these results.

Most of the presented data reproduce previously published findings by Schluter et al. (MBoC, 2007; Figures 1-2), Suzuki et al. (J. Cell Biol., 2019; Figure 3), and Suzuki et al. (eLife, 2021; part of Figure 4). This high degree of overlap with prior work substantially diminishes the novelty of the study.

The claim that "Co-immunoprecipitation experiments suggested that at least a part of Mvp1 associates with retromer. This represents the first biochemical evidence for a joint function of Mvp1/SNX8 and retromer" is not convincingly supported. The co-immunoprecipitation data are weak and lack appropriate positive and negative controls. The observed interaction could easily be indirect, mediated by other membrane-associated components. Moreover, no experiment directly demonstrates that the proposed retromer-Mvp1 association has functional significance. At a minimum, a point mutation in Mvp1 or retromer that disrupts the interaction and leads to a specific recycling defect would be required to substantiate this model.

In addition to missing experimental controls, most microscopy and western blot experiments lack quantitative analysis, making it difficult to assess the validity and reproducibility of the conclusions.

The manuscript lacks novelty, sufficient experimental control, and quantitative rigor. Without functional evidence and a clear mechanistic insight, the study does not meet the publication criteria of Journal of Cell Science.

Reviewer 2

SUMMARY OF THE ADVANCE MADE IN THIS PAPER AND ITS POTENTIAL SIGNIFICANCE TO THE FIELD

Comments for the author

The study is an examination of the trafficking of the Vps68 protein in yeast. Together with Vps55, the Vps68 protein regulates the half-life of the Ste6 protein, possibly via interactions with the endosomally localised ESCRT complex. The authors report that the Vps68 protein is itself trafficked between the endosome and the Golgi complex by retromer but in concert with Mvp1 - the yeast SNX8 homologue.

This is something of an 'old school' study and as such is relatively easy to read, digest and embrace. It is refreshingly free of hype and hyperbole and reports findings that look novel and will be of interest to those studying endosomal protein sorting and the role of retromer in particular. There are areas where the manuscript can be improved and I feel that some additional experimental data will strengthen the conclusions. Set out below are the changes, edits and additions I feel are appropriate.

SUGGESTIONS TO AUTHORS

1. In the introduction, the wording should be modified to bring the text in line with findings over the last few years that indicate that the Vps35-29-26 trimer may play a structural role (by forming arch-like assemblies on endosomal membranes) and the Snx dimer (Vps5-17) could play a role in sorting cargo.
2. Last paragraph of Introduction, change 'depended' to dependent.
3. In figure 2, why does there appear to be green fluorescence in the vacuole for some of the constructs - notably the truncation constructs?
4. In figure 2, do the GFP-tagged versions of Vps55 and Vps68 localise to what has been termed the 'class E' compartment? For example, what does the localisation look like in a vps27 mutant?
5. The authors describe the cytoplasmic tail of Vps10p as 'short' even though it is approx 150 amino acids in length which many would consider to be quite long for a cytoplasmic tail that does not contain any obvious functional domains. The authors also fail to mention the FYVF motif in the Vps10p tail that plays an important role in regulating/mediating the trafficking of Vps10p (see PMID: 8534908). The FYVF motif lies in the region of the Vps10p tail that has been highlighted by the authors (1428-1495) that they deem critical for Vps10p localisation. Additionally, this motif is noteworthy because it bears some similarity to a motif in the cytoplasmic tail of the SorL1 protein - a Vps10p-domain containing protein involved in regulating APP localisation and processing (FYVFSN in Vps10p, FTAFAN in SorL1) (see PMID: 22279231). Thus, this section of the manuscript needs a bit of a rethink and some rewording.
6. Retromer-dependent recycling of Vps10p was described in Seaman et al., 1997; Seaman et al., 1998. These should be cited in addition to Suzuki et al., 2019.
7. Does the Vps10 construct that carries the sorting motif from Vps68 (as shown in figure 5) rescue CPY sorting in a vps10delta mutant?
8. The colP of Mvp1 with Vps26 (shown in Figure 7) would be enhanced if it could also be achieved using antibodies against Vps35 or perhaps Vps5p.
9. I'm unconvinced the exploration of the phosphorylation state of Vps55 and Vps68 really adds much to the story. In my view, this could be removed from the manuscript to sharpen the focus on the story centred around retromer/Mvp1p recycling of Vps68.
10. The examination of the functionality of the GFP-tagged versions of Vps55 and Vps68 (shown in figure S1) would possibly be better accomplished using CPY sorting assays as the differences in the half-life of Ste6p is fairly subtle.

Author response to reviewers' comments

Reviewer 1

Lack of novelty

I do not agree that "most of the presented data reproduce previously published findings". Part of the data extend the observations of Schluter et al. (2008) that Vps55 and Vps68 form a complex by showing that the phenotypes of the single mutations are not additive and by showing that the intracellular localization of the two proteins is dependent on each other. This is information not present in Schluter et al.

But the main focus of the study is the recycling of the Vps68 protein. This has not been examined before. The main finding of the study is that Vps68 contains a tyrosine-based recycling signal at its C-terminus, which is retromer dependent. This is a surprising finding in light of the results obtained with Vps55.

There is a discrepancy in the literature with respect to the sorting factor dependence of Vps55. Bean et al. (2017) claim that Vps55 can be recycled by a retromer dependent mechanism as well as

by an Mvp1 dependent mechanism, while Suzuki et al. (2021) claim that Vps55 recycling is exclusively dependent on Mvp1 and not on retromer. The data in this study help to resolve this issue by supporting the claim of Suzuki et al.

In any case, I find it puzzling that the two subunits of the same protein complex have a different sorting factor requirement. I would have expected that both proteins recycle together by the same mechanism. This conundrum still needs to be resolved.

Another peculiar finding is that tyrosine-based signals like the one identified at the C-terminus of Vps68 have been shown to bind to clathrin adapter complexes. It is an intriguing question whether clathrin and retromer functions are interconnected. This is discussed in the manuscript.

Further, evidence is presented that Mvp1 may physically interact with retromer. I agree that the co-IP efficiency of Mvp1 with Vps26 is low, much lower than the co-IP efficiency of Vps5 with Vps26, which I included for comparison. But the signal is specific and may thus be real. Since the co-IP efficiency of Mvp1 with Vps26 is not overwhelming, I softened all the statements about the Mvp1-retromer interaction throughout the manuscript, especially in the abstract.

Lack of experimental control and quantitative rigor

I don't know, where you see lack of experimental control. All experiments were performed carefully and were of course replicated. The manuscript at hand is a summary of five years of research. During that time many hypotheses were tested and the constructs were thoroughly studied in all sorts of combinations. I am confident about the presented results.

The results were quantified where appropriate. For instance, I quantified the number of endosomal dots and vacuolar rings for the different tagged variants of Vps55 and Vps68. But for most of the experiments the results were clear and did not require quantification. For instance, for the Vps10 localization experiments it is "all or nothing", endosomal dots or vacuolar rings. There is no need for quantification.

Reviewer 2

- 1) The introduction was amended.
- 2) The mistake was corrected.
- 3) As discussed in the manuscript, I think that the orphaned proteins (Vps55 or Vps68 without its partner protein or Vps68 without its recycling signal) become a substrate for the MVB-pathway and are degraded in the vacuole. Thus, the vacuolar lumen staining.
- 4) In the ESCRT-III mutants $\Delta vps20$ or $\Delta vps60$ sfGFP-Vps68 localizes to the class E dot. Since Vps68 localizes to endosomes as shown by Schluter et al. and by us, such a localization would have been expected. I haven't explicitly tested the $\Delta vps27$ mutant, but the staining should look like the staining in the ESCRT-III mutants.
- 5) A paragraph addressing this point was added to the manuscript.
- 6) The citations were included.
- 7) A new figure addressing this point was added (Fig. 5). The Vps68 signal completely restores CPY sorting to the Vps10 Δ variant.
- 8) The co-IP between Vps26 and Vps5 is included in the manuscript (Fig. 8B). The co-IP efficiency is indeed much higher than in the case of Vps26 and Mvp1. Since the significance of our Mvp1 co-IP is not clear at present, I have softened all statements concerning the Mvp1-retromer interaction.
- 9) The phosphorylation data were deleted from the manuscript.
- 10) This would require a CPY pulse-chase experiment. Because we don't have an isotope lab in our institute, I could not do this experiment. I tried to address this issue by a gal-depletion experiment with *GALp-PRC1*, but the results were inconclusive, due to the presence of a massive m-CPY band and due to the fact that p2-CPY is formed from p1-CPY and consumed by conversion to m-CPY at the same time. Although the effects of the mutants on Ste6 turnover are moderate, they are reproducible and thus reliable.

Second decision letter (sister journal)

MS Title: Recycling of Vps68 depends on retromer and Mvp1/SNX8

Author: Ralf Kölling

Many thanks for submitting your revised manuscript to us.

We have now reached a decision. The revised manuscript has now been examined by a third expert reviewer, who provided an independent evaluation of the revised manuscript and rebuttal as I promised, and I have also sought independent evaluation of the rebuttal and revision from another editor. As I stated in my decision letter of October 22nd, the revised manuscript would need to address the technical concerns but also the question of novelty. Unfortunately, after these consultations, we have decided that your revised manuscript does not provide sufficient advance for publication in our journal and I am sorry to inform you that we cannot consider it further.

I am very sorry to give you such disappointing news, but it takes a very enthusiastic recommendation by the referees for a manuscript to be accepted.

Transfer to Biology OpenFirst decision letter

MS ID#: bio.062518

MS Title: Recycling of Vps68 depends on retromer and Mvp1/SNX8

Author: Ralf Kölling

I have now reached a decision on the above manuscript.

The reviewer reports are shown at the bottom of this email.

As you will see, the reviewers gave favourable reports, but raised a few points that will require amendments to your manuscript. I hope that you will be able to carry these out, because we would like to be able to accept your paper. If it is possible to carry out the additional experiment proposed by Reviewer 1, that would be ideal, but if you do not agree with this suggestion, please explain clearly why this is so.

At this stage, we also ask you to ensure your manuscript complies with our formatting guidelines - please see our manuscript preparation guidelines for details. Provided you are able to fully address the referees' comments, we are positive about publication of your paper (we accept over 95% of revision submissions) and therefore hope you won't mind any extra work involved in reformatting your manuscript at this point.

Please upload both a 'clean' version of your Word file, along with a highlighted version clearly showing where you have made changes in the revised manuscript. Please avoid using 'Track changes' in Word files as these are lost in PDF conversion.

I should be grateful if you would also provide a point-by-point response detailing how you have dealt with the points raised by the reviewers in the 'Response to Reviewers' box. If you do not agree with any of their criticisms or suggestions, please explain clearly why this is so.

Reviewer 1

Comments for the author

I have previously reviewed this manuscript for another journal. I was generally supportive and I note that the authors have responded to the comments/suggestions I made previously. There is only one additional suggestion I would make. If it is not possible for the authors to perform a metabolic-labelling experiment to monitor CPY sorting, then I would suggest an alternative: The secretion of CPY (in presumably the p2 form) could be monitored by growing the yeast to mid-log growth, converting to spheroplasts (using zymolyase) and then returning to culture for 30-60 mins. After this, the yeast can be centrifuged and culture medium collected and precipitated (using TCA perhaps). CPY can then be analysed by SDS-PAGE and western blotting. This method could enable an analysis of whether certain constructs etc can rescue the CPY sorting defects and thus give some clear indication as to the functionality of the respective construct.

Reviewer 2

Comments for the author

The manuscript by R. Kölling explores the recycling of the Vps55/Vps68 complex, involved in late endosome-to-vacuole transport. The author shows that the half-life of the Ste6 peptide transporter was increased in the absence of either Vps55 or Vps68, and that the phenotype of the double deletion mutant was the same as that of each single mutant. This result supports the conclusion that these two proteins form a functional complex. In addition, the stability of Vps55 and Vps68 was shown to be dependent on the presence of the other. The author identifies a tyrosine-based sorting signal in the cytosolic tail of Vps68, and demonstrates that this motif is required for the correct localization of the Vps55/Vps68 complex. Consistent with these results, recycling of Vps68 was shown to be dependent on retromer, in addition to Mvp1/SNX8. However, only Mvp1 is required for Vps55 recycling. Hence the two components of the Vps55/Vps68 complex have only one shared recycling pathway, with Vps68 in addition having a requirement for retromer for its recycling. This study is well done and I have only a few minor comments relating to the text.

Detailed comments:

1. Page 1, Abstract, lines 38-39

The co-IP signal, even though weak, appears to be specific. I suggest changing the final two sentences of the abstract from:

"In co-immunoprecipitation experiments a weak co-IP signal was detected between Mvp1 and the retromer subunit Vps26. Thus, there is a chance that Mvp1 could bind to retromer."

to:

"In co-immunoprecipitation experiments a weak co-IP signal was detected between Mvp1 and the retromer subunit Vps26, indicating a physical association between Mvp1 and retromer."

A summary sentence could be added to the end of the abstract.

2. Page 8, second paragraph of discussion, lines 35-36.

"But here for the first time we were able to detect a co-immunoprecipitation signal between Mvp1 and a retromer subunit. But since this signal is much weaker..."

Should be changed to

"Here, for the first time, we were able to detect a co-immunoprecipitation signal between Mvp1 and a retromer subunit. However, since this signal is much weaker..."

Reviewer's Responses to Questions

Experimental quality

Does each figure have the proper controls?

If 'No', please indicate reasons in Comments for Author box below.

Reviewer #1:

- Yes

Reviewer #2:

- Yes

Were the data analyzed using appropriate statistical tests?

If 'No', please indicate reasons in Comments for Author box below.

Reviewer #1:

- Yes

Reviewer #2:

- Yes

Reproducibility

Were experiments performed using adequate number of biological replicates?

If 'No', please indicate reasons in Comments for Author box below.

Reviewer #1:

- Yes

Reviewer #2:

- Yes

Does the methods section provide sufficient detail to permit reproducibility?

If 'No', please indicate reasons in Comments for Author box below.

Reviewer #1:

- Yes

Reviewer #2:

- Yes

Completeness

Are the manuscript's conclusions supported by the data?

If 'No', please indicate reasons in Comments for Author box below.

Reviewer #1:

- Yes

Reviewer #2:

- Yes

Scholarship

Do the authors cite and discuss the merits of data that would argue for and against their conclusion?

If 'No', please indicate reasons in Comments for Author box below.

Reviewer #1:

- Yes

Reviewer #2:

- Yes

Does the manuscript title & abstract accurately reflect the contents of the manuscript, without hyperbole?

If 'No', please indicate reasons in Comments for Author box below.

Reviewer #1:

- Yes

Reviewer #2:

- Yes

First revision

Author response to reviewers' comments

Response to reviewers

Reviewer #1

“Simulation of a pulse-chase experiment”. I doubt that we can extract meaningful results from such an experiment. There are a lot of problems associated with it. We got cell wall stress due to DTT, zymolyase and high pH treatment (pH 9.4!), we got osmotic stress due to high sorbitol and we got starvation conditions. And after all these stresses we have to bring the cells back to life. Although the effect of *VPS55* and *VPS68* mutants on Ste6 turnover is only moderate (2-fold stabilization), it is very reliable and was consistently observed over the years. So, I feel confident with the results. And since the point in question is not a major part of the story, I would rather not do this experiment.

Reviewer #2

The text was corrected as suggested.

Second decision letter

MS ID#: bio.062518R1

MS Title: Recycling of Vps68 depends on retromer and Mvp1/SNX8

Author: Ralf Kölling

I am happy to tell you that your manuscript has been accepted for publication in Biology Open, pending our standard publication integrity checks. It was accepted on 11th March 2026.